# Influenza B Virus Vaccine Innovation through Computational Design

**DOI:** 10.3390/pathogens13090755

**Published:** 2024-09-02

**Authors:** Matthew J. Pekarek, Eric A. Weaver

**Affiliations:** Nebraska Center for Virology, School of Biological Sciences, University of Nebraska-Lincoln, Lincoln, NE 68583, USA; mpekarek2@huskers.unl.edu

**Keywords:** influenza B viruses, viral evolution, surveillance, vaccines, computational design, immunogen design, SARS-CoV-2

## Abstract

As respiratory pathogens, influenza B viruses (IBVs) cause a significant socioeconomic burden each year. Vaccine and antiviral development for influenza viruses has historically viewed IBVs as a secondary concern to influenza A viruses (IAVs) due to their lack of animal reservoirs compared to IAVs. However, prior to the global spread of SARS-CoV-2, the seasonal epidemics caused by IBVs were becoming less predictable and inducing more severe disease, especially in high-risk populations. Globally, researchers have begun to recognize the need for improved prevention strategies for IBVs as a primary concern. This review discusses what is known about IBV evolutionary patterns and the effect of the spread of SARS-CoV-2 on these patterns. We also analyze recent advancements in the development of novel vaccines tested against IBVs, highlighting the promise of computational vaccine design strategies when used to target both IBVs and IAVs and explain why these novel strategies can be employed to improve the effectiveness of IBV vaccines.

## 1. Introduction

Prior to the global spread of SARS-CoV-2, influenza viruses were one of the primary causes of infectious respiratory disease every year [1,2]. Up to 650 thousand deaths worldwide were directly attributed to influenza infection [3], and an estimate found that up to 11% of the United States (US) population was typically infected each year [4]. Severe influenza infections are caused by the circulation of seasonal influenza viruses that are classified as type A (IAVs) and type B (IBVs) [3]. Influenza viruses are classified based on the sequence and antigenicity of their surface glycoproteins, hemagglutinin (HA) and neuraminidase (NA). HA is often the target of vaccine development [5,6,7,8,9], while NA has historically been a major target of influenza antivirals [10,11,12,13,14]. IBVs are separated into two lineages, B/Victoria-like and B/Yamagata-like, based on HA antigenic differences and named after the respective reference strains, B/Victoria/2/1987 and B/Yamagata/16/1988. Both lineages have co-circulated in the global population since at least the early 1980s [15]. 

While licensed vaccines and antivirals are available, the efficacy of both can be limited due to the constant mutation and evolution of circulating influenza strains. Mutations and genetic diversity are principal obstacles in producing a commercial vaccine that protects well against divergent IBV strains in a single formulation. Antigenic drift, or the accumulation of point mutations in the surface glycoproteins [16,17], can lead to the evasion of vaccine-induced immunity in HA [8,18,19] and can be a key component in driving antiviral resistance in NA [20,21]. To combat this obstacle, novel methods of designing influenza vaccines and antivirals continue to improve the induction of cross-reactive immune responses that inhibit viral replication in order to tackle the genetic diversity inherent to influenza virus evolution. 

Historically, vaccine and antiviral research for IAVs has been prioritized over IBVs. Although some IBV infections have been reported through the surveillance of marine mammals and swine, the viruses found in non-human hosts are antigenically similar to circulating human strains isolated around the same time [22,23,24], suggesting that these animals may be susceptible to IBV infection but not support persistent circulation. This is drastically different from IAVs that have many known animal reservoirs [25]. Therefore, IBVs are not considered to have a high probability of causing a potential pandemic [26]. This has driven much of the influenza research to focus on IAVs for pandemic preparedness strategies. However, recent studies have highlighted the significance of IBV infection on patients [27,28,29], especially those in high-risk categories, such as children [30,31,32] and the elderly [33]. Additionally, the 2019–2020 influenza season saw the emergence of a new clade of IBVs, which was associated with a disparate circulation pattern compared to typical flu seasons [34]. This evidence suggests that IBVs can cause significant disease burden and warrants dedicated research attention similar to that for IAVs.

The emergence of SARS-CoV-2 (COVID-19) from Wuhan in late 2019 [35] and the non-pharmaceutical measures implemented to lessen the effects of the pandemic altered the circulation patterns of influenza and other respiratory viruses [36,37,38,39]. Despite the continued circulation of SARS-CoV-2, the relaxation of these measures in most areas has led to the circulation of respiratory viruses at, or close to, levels seen prior to the pandemic [40,41,42,43]. However, the impacts of the change in IBV circulation patterns are still seen through global surveillance data [44]. The resurgence of influenza and other respiratory virus infections signals the need to maintain focus on influenza viruses as significant respiratory viruses that will continue to circulate along with SARS-CoV-2. Additionally, the prevention or mitigation of severe influenza disease may prove crucial for successful health system management as we continue to handle COVID-19 and other severe infections.

Due to the severity of disease and the likelihood of continued circulation, it is important to understand current advances in the IBV vaccine field. In this review, we highlight key challenges in the evolutionary patterns of IBVs that influence vaccine development. In addition, we analyze recent pre-clinical advancements in IBV vaccine design and delivery. Finally, we discuss the potential for computational immunogen design strategies and methods targeting IBVs based on previous successes targeting IAVs and how these strategies could improve the effectiveness of current influenza vaccines.

## 2. Evolutionary Patterns and Recent Surveillance

As human-specific viruses, the evolutionary patterns of IBVs are distinct from IAVs. The first identified case of an IBV was described in 1940 [45]. In the late 1980s, the circulation of two distinct antigenic lineages was discovered based on the reactivity of ferret antisera [15] and later differentiated by HA sequence variation [46]. A recent report by Rosu et al. found that mutations near the receptor-binding site of HA are responsible for major antigenic changes associated with lineage divergence. When site-directed mutations were induced near the receptor-binding site, antisera reactivity resembled the heterologous lineage that the mutations were directed toward rather than the remaining backbone of the homologous lineage [47]. Geographic and temporal infection patterns between the two lineages throughout the 1990s continued to drive the evolution of each lineage until global circulation of both lineages throughout the same season began in the 2000s [48,49]. The inter-lineage reassortment of internal genes has been described extensively [48,50,51,52], but no B-HA reassortment has been described to date. This suggests that B-HA antigenic drift is a crucial evolutionary mechanism for IBVs to avoid neutralizing antibodies in the population. The antigenic drift of B-HA primarily occurs through a buildup of point mutations, leading to the evasion of host memory immune recognition. Fitness pressure on the viruses preferentially selects for variation in the major B-HA antigenic regions that avoid immune neutralization [53]. These sites are found in the HA1 domain of the protein and consist of the 120-loop, 150-loop, 160-loop, and 190-helix [54]. Neutralizing antibodies against other regions of HA can also develop. However, these sites are typically targeted after an increase in age and, likely, multiple IBV exposures [55]. B-HA was found to readily tolerate mutations in the HA head [56], supporting the ability to avoid immune recognition through these point mutations. Despite the reliance on antigenic drift for the evasion of neutralizing antibodies, the rate of nonsynonymous mutations in HA of IBVs is slower than that of IAVs [52,57,58]. Hypotheses proposed to address this include less errant polymerase activity, lower immunogenicity, and evolutionary constraints on tolerable mutations in HA [57,58,59,60]. These factors set the stage for what the viruses must overcome to maintain the annual global epidemics observed every flu season (Figure 1).

Signs of IBVs overcoming these hurdles were seen at the start of the 2019–2020 flu season in the Northern Hemisphere before the spread of SARS-CoV-2. The Global Influenza B Study [61] found that from 2000 to 2018, in 31 countries globally, IBVs were responsible for around 23% of all influenza cases. These seasons varied in severity, with IBVs predominating in all cases around once every seven seasons. Peak epidemic activity was around one month after the peak circulation of IAVs [62]. However, a dramatic shift in epidemiology was observed early in the 2019–2020 season. The emergence and takeover of a new B/Victoria-lineage (B/Vic) subclade in the United States occurred after a high prevalence of B/Yamagata-lineage (B/Yam) circulation was observed in the three seasons prior. This was representative of the temporal circulation commonly observed with IBVs. Waning immunity to B/Vic HA, combined with the emergence of a large antigenic shift in B/Vic HA (characterized by a three amino acid deletion in the 160-loop of B-HA), led to the early onset and higher peak epidemic activity of IBVs in the US compared to prior seasons [34]. These viruses then dominated worldwide IBV circulation [63,64,65,66]. 

**Figure 1 pathogens-13-00755-f001:**
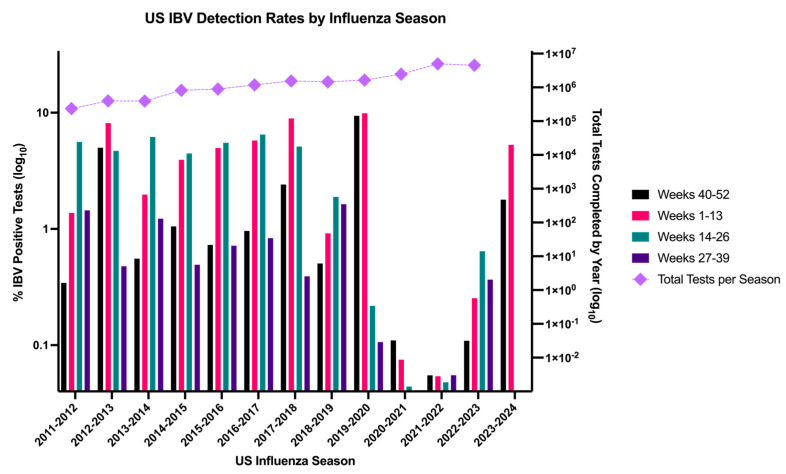
Seasonal patterns of IBV circulation and testing in the US. Surveillance data from the US CDC Fluview^®^ Interactive over the previous 13 influenza seasons from 2011–2012 to 2023–2024. Seasons were broken down into 4 quarters from the start of the US influenza season (week 40 of the calendar year) to best observe intraseasonal circulation patterns. A percent positivity rate was calculated by dividing the number of IBV positive tests within each quarter by the total number of tests completed in the quarter and is plotted on the left y-axis. Data were combined from public health and clinical labs. The total number of tests completed for each season is plotted on the right y-axis. Only data from completed quarters were included for the 2023–2024 US flu season. All data were obtained from [67].

However, by mid-2020, many countries had adopted social distancing and/or isolation measures to limit the spread of SARS-CoV-2. Additionally, masks were adopted in most regions to limit the spread of the virus through respiratory droplets. Unsurprisingly, these measures to limit the spread of one respiratory virus also affected the spread of IBVs [36,37]. Data collected from the Centers for Disease Control in the US show a relatively stable pattern of positive test rates up until the spring of 2020, where the percent positivity began to sharply decline despite the number of tests increasing (Figure 1) [67]. This was not limited to IBVs. Takeuchi and Kawashima report the effects of mask use on the reduction of influenza in the Southern Hemisphere during the initial stages of the COVID-19 pandemic [39]. As more time passes since the initial COVID-19 pandemic response, and non-pharmaceutical measures to prevent respiratory virus spread are no longer prevalent, it appears that the total percentage of IBV infections in the US is reverting to pre-pandemic circulation patterns (Figure 1). However, this increase in total cases does not signal a return to pre-pandemic IBV circulation patterns. A significant development during the COVID-19 pandemic was the reversion from lineage co-circulation to only B/Vic detection. Researchers have postulated that the B/Yam lineage has been extinct since the initial outbreak of SARS-CoV-2 [44,68]. Further, this has led to recommendations to revert vaccine formulations to those only containing B/Vic lineage HA in commercial influenza vaccines [69] and raised the possibility of the eradication of influenza B viruses [70]. The possibility that the lineage continues to circulate undetected does remain. However, as more time passes without a re-emergence in the surveillance, the likelihood of this possibility decreases dramatically. This will remain a challenge to current vaccine formulation and future vaccine development directed toward IBVs until a clearer picture comes into focus.

## 3. The Current State of Clinical IBV Vaccines and Trends in Immunogen Design for Pre-Clinical Vaccine Research

Historically, commercial influenza vaccines contained strains from the H1N1 and H3N2 IAV subtypes and a singular IBV strain from either the B/Vic or B/Yam lineage in a trivalent formulation. However, vaccines only containing a single IBV lineage often led to vaccine mismatches globally when the lineage found in the formulation was not the predominantly circulating lineage that season [71,72,73,74]. The current standard of vaccination in the United States is a quadrivalent inactivated vaccine (QIV) containing strains against both antigenic lineages of IBVs as well as H1 and H3 IAVs [75]. Issues with the current influenza vaccines are well known. The manufacturing time of the vaccine requires strains to be selected months in advance of batch distribution. This leaves time for antigenic drift from the strain incorporated in the vaccine to occur in circulating strains and leads to mismatches during the influenza season. When analyzing QIV vaccine effectiveness in the United States, mismatches between either the IBV or IAV vaccine component(s) with circulating strains throughout influenza seasons have been well-documented [76,77,78]. Another issue with current vaccines is that the inactivated virus platform typically induces immune responses that are strain-specific with little cross-reactivity [79]. This leads to many breakthrough infections in people who received the vaccine, though the disease severity may still be lessened. Finally, experts agree that the production of vaccines using embryonated eggs likely leads to egg-adapted mutations, which may limit vaccine effectiveness [80]. Due to these current limitations, increased attention has been placed on improving the current methods of vaccinating against influenza.

The goal of a “universal flu vaccine” has long been pursued, but within the last decade, dedicated efforts to achieving this have increased drastically. While more effort has been invested into developing universal IAV vaccines, universal IBV vaccines may be a more achievable target due to the host and genetic diversity constraints not faced by IAVs [81]. Here, we will briefly focus on the experimental immunogen designs and alternative immunogen strategies that have been tested for improving vaccine effectiveness against IBVs in pre-clinical studies. Tsybalova et al. recently provided an update on epitope-targeting and universal IBV vaccine candidates ranging from pre-clinical studies to clinical trials [9]. While there is significant effort to develop vaccines that target cross-reactive antibodies for prophylaxis, recently published clinical research for IBV vaccines has mostly focused on testing currently licensed vaccines, comparing different production and delivery methods, and characterizing their impact on vaccine efficacy [82,83,84,85]. Therefore, new IBV vaccine designs reaching clinical trials are essential for improving influenza vaccine effectiveness.

Much effort has been put into identifying the role HA stalk-directed immunity can have on influenza vaccine effectiveness. As this region is less tolerant to mutations than the head region [56,86], these stalk-directed vaccines lead to less strain-specific immune response induction [17] by directing immunity away from antigenic sites in the HA head (Figure 2). One way to direct immunity toward the stalk is through sequential immunization with “chimeric” HA proteins. To achieve this, multiple immunizations are administered with each dose containing identical stalk domains, while a unique head domain is delivered in each dose. This leads to a naïve response to the immunodominant head region, while a memory response against the stalk is boosted. This design has been widely described over the last decade, including many reports specifically targeting the IBV stalk region. First described in 2013 to target the IAV H1 stalk [87], researchers adapted this strategy to the stalk of B-HA. They produced two different recombinant protein forms varying only in the amino acids linking the head region to the stalk region to compare the optimal method of tethering an IAV HA head to the B-HA stalk. BALB/c mice were completely protected from lethal B/Vic, B/Yam, and B/Lee/1940 (ancestral IBV strain) challenge following the immunization series with the optimal vaccine design. Protection was likely mediated through antibodies to the immunogen despite the lack of a high neutralizing antibody titer. The evidence suggests that antibody-dependent cellular cytotoxicity (ADCC) likely played a role in protection [88]. 

The strategy to boost stalk-directed immunity was further refined from chimeric vaccination. Rather than replace the entire B-HA head, the antigenic regions of the B-HA head were replaced with multiple unrelated IAV antigenic sequences. These recombinant B-HA proteins are referred to as “mosaic” B-HAs. Sequential vaccination with a mixed DNA/protein platform again led to protection from both B/Vic and B/Yam lethal challenge, with evidence suggesting that non-neutralizing antibodies were crucial in protection from challenge [89]. Further, the “mosaic” B-HAs were expressed in a whole inactivated virus (WIV) platform to mimic the current standard of immunization. When adjuvanted, the mosaic B-HA WIV vaccine induced stalk-specific antibody responses that were protective from heterologous lineage challenge and durable for over 40 weeks after prime immunization. Again, ADCC-stimulating antibodies were detected and likely played a role in protection even nearly a year after immunization [90]. 

Another stalk-directed vaccine strategy has also been recently tested. Produced as a recombinant protein, a modified B-HA stalk protein was cross-linked onto an IBV nucleoprotein (B-NP) core and delivered as a protein nanoparticle [91]. This nanoparticle vaccine induced class-switched IgG antibodies directed toward the B-HA stalk and modest cell-mediated immune responses. After vaccination, mice were protected from lethal challenge with strains from both the B/Vic and B/Yam lineages. With the previously discussed stalk-directed strategies, these results support further investigation into stalk-directed IBV vaccines to continue to improve their pre-clinical promise. However, little other work in structural immunogen design for IBV vaccines is currently being pursued. Therefore, it remains unclear how further immunogen designs, like B-HA stalk vaccines, could impact IBV vaccine effectiveness. 

Immunogen design is not the only prominent area of research for IBV vaccines. Multivalent influenza vaccines are also emerging as a promising idea for IBV-targeting. Delivering multiple immunogens in a single vaccine increases the potential to develop a broad immune response rather than delivering a single target to the host immune system. Typical seasonal vaccines include one B/Vic and one B/Yam B-HA to induce lineage-specific immune responses. Cardenas-Garcia et al. reviewed studies showing the efficacy of different reverse genetics systems for the development of IBV vaccines, including multivalent B-HA deliveries and combinations with various IAV proteins [92]. Published more recently, a study tested a multivalent mRNA vaccine containing an HA gene from all 20 known HAs from IAVs and IBVs (H1-H18 IAV subtypes, B/Vic and B/Yam lineages) delivered in a lipid nanoparticle carrier (mRNA-LNP vaccine). Despite having 20 different immunogens delivered, multivalent mRNA-LNP vaccination led to detectable antibodies against each IBV lineage, though the data from challenge experiments using IBV strains were not reported [93]. Another study reported exciting results by delivering a pentavalent mRNA-LNP formulation designed based on lineage-specific B-HA, B/Colorado/6/2017 (CO/17) IBV neuraminidase (B-NA), B-NP, and matrix protein 2 (B-M2). Intradermal delivery either as a single antigen or as a multivalent vaccine induced both neutralizing and non-neutralizing antibody responses in mice. Both CD4^+^ and CD8^+^ functional T cells were detected, and the vaccine protected against morbidity and mortality after challenge even at a low (50 ng) dose administered in a single immunization [94]. While new immunogens were not designed in these reports, studies such as these may drive the IBV vaccine field toward novel ways to deliver multivalent vaccines with less concern over the number of immunogens delivered dampening specific immune responses. Further, this multivalent delivery strategy may be more effective at inducing both antibody and cell-mediated immune responses to improve protection from vaccination compared to current inactivated vaccines.

Targeting IBV NA (B-NA) in vaccine research has recently gained traction as an alternative immunogen to B-HA [95,96,97,98]. A recent study showed that anti-B-NA serum can protect mice from homologous lineage challenge and improve inter-lineage protection compared to anti-B-HA serum. [97]. In a separate study, the intranasal delivery of recombinant B-NA limited the transmission of IBVs in a guinea pig model [99]. This evidence suggests that B-NA as a vaccine target could induce protective immunity similar to vaccines targeting B-HA. However, B-NA as a vaccine immunogen is understudied compared to IAV NA vaccine designs, where much more is known about anti-NA immunity [98,100,101,102,103]. Therefore, further studies supporting the protective efficacy of B-NA as a primary immunogen are needed. 

Among the internal IBV proteins, only B-NP has received much attention as a vaccine target, mostly due to the induction of strong CD8^+^ T-cell responses. The intranasal delivery of replication-defective adenovirus (rdAd)-vectored B-NP induced strong antigen-specific, tetramer^+^ CD8^+^ T-cell responses, which localized to the lungs after challenge. A single immunization with the rdAd-B-NP vaccine protected mice from a lethal challenge at the peak immune response [104]. A DNA vaccine delivering a single B-NP was also shown to protect mice from lethal challenge after the induction of T-cell migration into the lungs [105]. However, this vaccine did not protect the mice from challenge without an outside stimulus to induce T-cell migration to the lungs. Moving forward, vaccine development targeting other internal proteins, such as the IBV polymerase complex, could receive some attention based on results from IAV vaccines targeting polymerase proteins [106,107,108]. However, the efficacy of polymerase or other internal protein-targeting vaccines may be limited due to the low induction of effective antibody responses despite robust T-cell responses to vaccination. Altogether, the IBV vaccine research field is beginning to catch up with IAV vaccine development through dedicated studies aiming at maximizing anti-IBV immunity in animal models. Additionally, the advances of IBV vaccines to clinical trials could further advance the standard of protection against IBVs [9]. It remains to be seen what effect some of these new research directions may have on future IBV vaccine research in addition to total influenza and other viral vaccine development.

## 4. Computational Design Strategies for Influenza Vaccines

One area of IBV vaccine research that is still currently lagging behind IAV research is the use of computational immunogen design strategies. These strategies all utilize computational algorithms or workflow to incorporate large populations of sequence data into the design rather than choosing one or more wild-type sequences. This allows a vaccine design to incorporate the sequence diversity present in populations targeted for protection. For computational designs, large amounts of phylogenetic information are necessary to design effective vaccine immunogens. Table 1 provides information on computational design strategies used for IBV vaccines as well as a description of other computational algorithms used to test IAV vaccines.

### 4.1. Single-Target Consensus Designs

The most basic form of computational vaccine designs is a consensus sequence immunogen. There are many ways to design consensus immunogens, ranging from simple consensuses of unique sequences at each amino acid position to multi-level consensus building. The workflow and effectiveness of these consensus designs for influenza vaccines have previously been reviewed [8]. Focusing on consensus vaccines targeting IBVs, a simple consensus vaccine using unique B-HA stalk sequences was delivered using an adenoviral vector. The full-length consensus B-HA stalk vaccine induced strong antigen-specific T cells and antibodies inducing ADCC. Notably, when delivered intranasally, this vaccine was able to fully protect mice from lethal challenge with strains from either the B/Vic or B/Yam lineage [109]. This currently remains the only simple consensus vaccine design reported to specifically target IBVs, though there are some notable drawbacks to this strategy.

Most simple consensus vaccine designs face a common issue: biased population data toward more recent sequences due to increased sequencing capacity and better repositories developed over time. IAV consensus vaccine design studies have designed multiple ways to address this potential biasing issue. Centralized consensus immunogens base the sequence design on representative sequences spread throughout the entire population rather than all sequences present in that population. These immunogens often minimize the antigenic distances between sequences within the population and localize near the center of the phylogenetic tree. To date, centralized consensus HAs have been designed and tested against H1 [110], H3, and H5 [111], and most recently, H2 [113] IAV. Delivered by a rdAd vector, these vaccines induced neutralizing antibody titers above a protective titer of 40, recalled T-cell responses significantly more robust than commercial comparator vaccines, and protected against lethal challenge from homosubtypic IAV strains in the mouse model [110,111,112,113]. 

Population diversity can also be addressed in consensus design by stacking multiple layers of consensus sequences to create a consensus of consensuses. This has led to the creation of a large set of computationally optimized, broadly reactive antigen (COBRA) vaccines. First developed as a strategy to protect against the H5N1 avian IAV [114], COBRA vaccines have since been expanded to target H1 [116], H2 [120], and H3 [117] as well as most recently the IAV N1 protein [124]. These vaccines have shown promise in a variety of animal models of influenza infection while also showing effectiveness in pre-immune models of vaccination. A summary of the COBRA immunogen designs described above and tested for other objectives can be found in Table 1. Both centralized consensus vaccines and COBRA vaccines are shown to induce a much wider breadth of protection than commercial vaccines or wild-type comparator sequences. 

### 4.2. Multivalent Computational Algorithm Designs

Computational design strategies are not limited to solely designing consensus sequences. Computational algorithms have been developed to expand on the success of consensus vaccines. One advantage that these computational design algorithms have over consensus strategies is the ability of the algorithms to generate multivalent vaccines targeting a single protein from the same input population while maintaining the putative functional regions. Multivalent consensus designs using the same input population may lose such functional sites by incorporating rare amino acids at highly conserved areas of the protein, disrupting protein function. Another advantage is the ability to incorporate immune response modeling into the algorithms to design optimized sequences. Consensus sequences are often designed with one amino acid, leaving the design vulnerable to the disruption of epitopes readily recognized by the immune system. Therefore, computational algorithms may enhance the immunogenicity of vaccines when compared to consensus immunogens. 

One such algorithm is the Mosaic Vaccine Design algorithm. Originally used to induce broadly cross-reactive immune responses to HIV infection, the Mosaic algorithm designs immunogen sequences by simulating recombination events between sequences within an input population to maximize potential T-cell epitopes (PTEs). Then, the initial progeny goes through additional rounds of recombination to simulate whether the newly recombined sequence increases the coverage of PTEs found in the population. Further rounds of optimization are conducted before the algorithm finally provides the optimal vaccine sequence(s) [135]. The algorithm provides researchers with the ability to choose the number of immunogens for the algorithm to design as well as the size of the recombination regions to use in the process. After showing promise in inducing robust T-cell mediated immune responses in non-human primate models of HIV vaccination [136,137], a mosaic H5 influenza vaccine cloned into a modified Vaccinia Ankara (MVA) viral vector showed homosubtypic protection in mouse and non-human primates [128,129] through decreased viral replication in the lungs and neutralizing antibody induction. MVA H5 Mosaic (H5M) induced robust hemagglutination inhibiting (HAI) antibodies against multiple avian H5 influenza clades and protected from divergent avian H5 influenza in the mouse model. This protection was shown to be durable up to 6 months post-vaccination. Interestingly, the researchers seemed to also show some cross-reactivity with human seasonal H1N1 induced by their H5M vaccination. Peptide library screening of BALB/c splenocytes showed robust reactivity to H5 peptides along the whole length of the protein. However, some induction of IFN-γ^+^ splenocytes was detected near the C-terminus of an H1 peptide library, which was significantly greater than vaccination with a wild-type H5 encoded in an MVA vector [128]. In non-human primates, challenge with a pdm09 H1N1 virus led to detectable T-cell responses through an IFN-γ ELISpot assay after vaccination with the H5N1 vaccine, which also suggests a role for heterosubtypic recognition after Mosaic vaccination [129]. Another study reported the effectiveness of a Mosaic H1 vaccine design. The vaccine induced cross-reactive neutralizing antibody and T-cell responses against strains from both the seasonal H1N1 and pdm09 H1N1 viruses. The vaccine, delivered in an rdAd vector, also protected mice from challenge with seasonal H1N1 viruses [130]. Together, these results show promise for computational algorithm-designed influenza vaccine immunogens.

Another computational algorithm currently used to test pre-clinical influenza vaccines is the Epigraph Vaccine Design algorithm. Created to streamline some aspects of the Mosaic algorithm, the Epigraph algorithm again works to maximize PTEs in the immunogen design. Rather than simulating potential recombination events, the Epigraph algorithm calculates the k-mer epitope frequency along the full length of the protein sequences within the population, moving one amino acid at a time. As the algorithm moves further along the protein sequence, it traces a path of the highest frequency contiguous epitopes and incorporates them into the output immunogen. To create a multivalent design, the algorithm then removes those epitopes and incorporates the second most frequent epitopes into the next immunogen and reiterates the process for each individual immunogen specified.

Originally used to design a therapeutic HIV vaccine and test a pan-Filovirus vaccine [138,139], Epigraph vaccines have since been tested for both swine and human H3 IAV. When delivered in an HAd5 vector, the swine H3 (swH3) Epigraph vaccine induced robust and broadly cross-reactive antibody and T-cell responses in mice. Significantly, these cross-reactive responses led to the protection of mice from a lethal human H3 (huH3) challenge despite little to no HAI antibody response being detected. This study also expanded into a target animal and detected robust, cross-reactive antibody and T-cell responses when used to vaccinate swine [131]. Further testing of the swH3 Epigraph vaccine in pigs showed not only robust activation of neutralizing antibodies and T cells after vaccination but also that the responses induced were durable. Longitudinal sampling of the pigs over 6 months showed HI titers remaining above a protective threshold for a majority of the clades tested despite the clinical vaccine waning to below protective, or even detectable, levels for many of the clades tested. This protection was further characterized through experimental infection 6 months post-vaccination. swH3 Epigraph vaccination limited infectious virus and induced much more robust recall T-cell responses in the lungs 5 days post-challenge [133]. The huH3 Epigraph vaccine also detected robust humoral and cellular immune responses in the mouse model. The T-cell response induced by vaccination is shown to be valuable for protection against lethal challenge through depletion studies. Further, when delivered in the ferret model, the huH3 Epigraph vaccine significantly outperformed a commercial vaccine comparator in the breadth of cross-reactive antibody responses and protection from viral challenge [132]. Based on the results of these studies, the Epigraph vaccine platform shows impressive promise as a way to better design multivalent vaccines to protect against a diverse virus population.

The pre-clinical success of computationally designed vaccine immunogens, including centralized and COBRA consensus vaccines and Mosaic and Epigraph algorithm vaccines, targeting IAVs isolated from various hosts and subtypes suggests that these strategies could lead to potential success when used to target IBVs as an input population. Not only have these methods consistently delivered similar results in the breadth of cross-protection and robustness of response but they have been replicated in many pre-clinical models of influenza infection. Recently, the first signs of potential for these immunogen design strategies targeting IBV vaccines have emerged in the literature. In ferrets pre-exposed to IBVs, COBRA B-HA vaccination induced neutralizing antibodies and protected ferrets from signs of clinical infection [127]. Further, a recent report detailing the immune response to a multivalent Epigraph B-HA vaccine showed signs of both neutralizing antibodies and IFN-γ^+^ T cells after vaccination in the mouse model. This vaccine also showed protection against a high-dose lethal challenge from mouse-adapted IBVs [134]. Based on these initial reports, it is likely that further investigation and optimization of computationally designed IBV vaccines will result in robust responses to vaccination and improve current vaccine effectiveness levels.

## 5. Concluding Remarks and Future Research Perspectives

Influenza B viruses have historically circulated following unique evolutionary patterns compared to the different subtypes of IAVs. Studies over the last decade have shown increased overall effectiveness when strains from both IBV lineages are included in a quadrivalent vaccine formulation [140,141,142]. However, the best path forward for influenza IBV strain incorporation into vaccine formulations is not clear. Therefore, further breakthroughs to increase the strength and duration of immune responses to IBV vaccination will ease questions about what strains to incorporate. Additionally, the vaccine delivery platform will play a role in determining the route of administration that ensures the optimal immune response is elicited and the clinical guidance is maintained [143,144]. To date, computationally designed influenza vaccines have primarily been delivered via intramuscular injection. To further uncover the potential of computational influenza immunogen designs, more studies comparing vaccines delivered via an intranasal route to an intramuscular route are required. Issues of effectiveness against IBVs are not restricted to licensed vaccines. Zaraket and colleagues concluded in a recent review that oseltamivir and other NA inhibitor influenza antivirals generally have a lower efficacy against IBVs than IAVs, though resistance mutations emerge less commonly [29]. Despite these molecular, evolutionary, and clinical differences, only recently has the influenza vaccine research field begun focusing on IBVs as a unique target rather than just incorporating them into vaccines alongside IAVs.

The global spread of SARS-CoV-2 had a large impact on the spread and circulation of IBVs. Much of this impact occurred due to the implementation of non-pharmaceutical measures, such as masking and social distancing [37,39]. The most recent circulation of IBVs prior to the initial pandemic wave was already uncommon compared to what had occurred in previous seasons [34], and combined with non-pharmaceutical measures to address SARS-CoV-2, it appears likely that IBV circulation has been permanently altered. Current surveillance data from the US appear to show a resurgence to “normal” IBV circulation patterns; however, only the B/Vic lineage has been detected [67]. Therefore, we must remain diligent in innovating vaccines targeting IBVs to protect against current circulation while also pushing for increased cross-protection amidst the uncertain future of IBV circulation. 

Positive effects of the SARS-CoV-2 pandemic can be seen across vaccine research fields. The global response to the spread of SARS-CoV-2 led to the unprecedented development of new vaccine technologies to rapidly respond to the global effects of the pandemic [145]. This push led to general public acceptance of new vaccine technologies, such as nucleic acid-based [146] and viral-vectored vaccines [147]. Prior to the necessity for a SARS-CoV-2 vaccine, nucleic acid- or viral vector-based vaccines and therapies were clinically limited to treating rare genetic disorders and restricted in their target population by the price of the therapies [148]. However, now the scope of these technologies is rapidly expanding to a variety of vaccine targets due to their flexibility in delivering immunogens and relatively low time to production. This is crucial, especially for vaccines targeting a highly divergent virus type with global circulation. Despite the promise of computationally designed vaccines, it is unlikely that a single design or delivery system will completely eradicate IBVs. Rather, the vaccines will likely still need to be updated in response to major antigenic changes or be delivered with periodic boosters to maintain population immunity. Therefore, combining the new technologies accepted during the COVID-19 pandemic with computationally designed IBV vaccines may lead to more successful prevention of IBV infections. These strategies may help overcome the possible limitations of computational designs through an enhanced capability to implement required updates and increase production speed to maintain protection quicker than is currently possible.

Historically, many cutting-edge vaccine technologies have been developed to induce protection against viruses with genetically diverse circulating populations. These have included viruses such as HIV and influenza viruses and now SARS-CoV-2. Developing vaccines against these diverse viruses has required innovative strategies currently in development or, in the case of HIV, yet to be developed. One of the promising immunogen design strategies developed over the last decade has been the use of computational algorithms to incorporate large amounts of sequence variation data into the vaccine design rather than relying on the prediction of possible circulating strains. Computational vaccine designs have repeatedly shown promise when designed based on diverse IAV subtypes. IBVs are not as genetically diverse as IAVs; however, the same principles are likely to be successful when applied to IBV vaccines. The next step is to improve our understanding of the robustness, breadth, and duration of immune responses after vaccinating with immunogens designed with computational design principles targeting IBVs.

## Figures and Tables

**Figure 2 pathogens-13-00755-f002:**
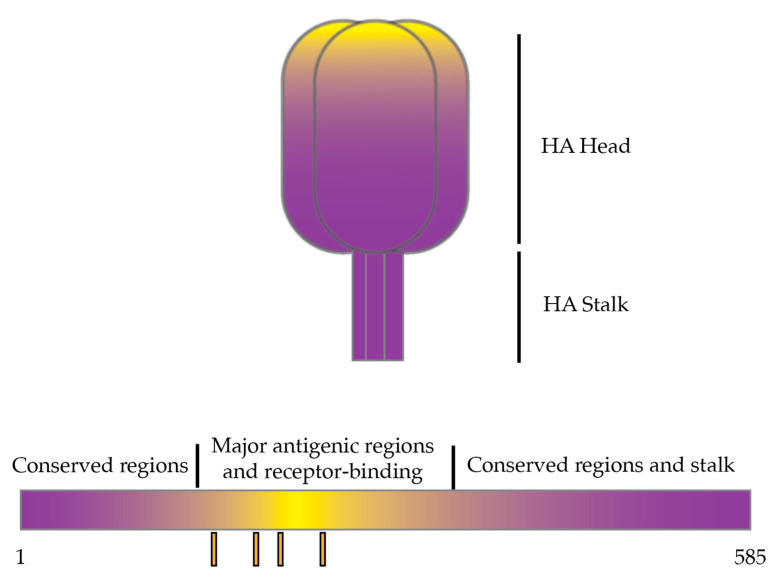
Schematic depicting objectives of stalk-directed vaccine strategies. Stalk-directed vaccines primarily direct immunity away from the antigenic and variable head region near the receptor-binding site [47,54] in favor of targeting more conserved regions of the protein near or in the HA stalk. The darker colors represent more immune direction toward that region. A three-dimensional representation of the HA trimer is shown above, and a schematic representing the linear amino acid sequence with conserved and antigenic regions (orange bars below) distinguished is also shown.

**Table 1 pathogens-13-00755-t001:** Summary of computational vaccine designs tested against influenza viruses. Targets, delivery platforms, animal models, and study results are summarized in the table.

Immunogen and Design	Influenza Target(s)	Vaccine Delivery Platform	Model	Study Highlights	References
Consensus IBV HA2	IBV	Recombinant Adenovirus	Mouse	Intranasal prime-boost delivery induced robust IgG antibody class-switching and protected mice from lethal challenge with B/Vic. Robust ADCC induced through vaccination. Fusion peptide and transmembrane domain critical for protection from lethal challenge.	[109]
Centralized consensus HA	H1, H2, H3, H5 IAV	Recombinant Adenovirus	Mouse	Immunization protected mice from multiple heterologous lethal challenge in a dose-dependent manner. Robust induction of IFN-γ^+^ splenocytes to greater levels than wild-type HA vaccines. Antibodies mediate HAI response against divergent strains within subtype. Multivalency led to increased neutralizing antibodies and IFN-γ^+^ splenocytes compared to commercial vaccines. Serotype switching of adenovirus vector leads to more robust HAI antibody responses in mice.	[110,111,112,113]
Computationally optimized, broadly reactive antigens (COBRA)	H1, H2, H3, H5, N1 IAV, IBV	Virus-like particle Recombinant/conjugated protein Split-inactivated virus particle Recombinant Marek’s disease virus/turkey Herpesvirus vector Cationic lipid nanoparticle	Mouse Ferret Non-human Primate Chicken	COBRA-designed final consensus layer outperforms clade-/subtype-specific vaccine designs in homologous and heterologous antibody production while maintaining critical protein structures. HAI antibodies induced against multiple clades of viruses using homologous subtype COBRA immunization. Heterologous immunogen boosting or cocktail immunization led to increased cross-reactivity and robustness of HAI antibodies and protection from challenge compared to wild-type immunogens. Swine H1 COBRA induces HAI antibodies against both swine and human H1 strains. Nanoparticle delivery of COBRA immunization boosts T-cell response compared to VLP delivery.	[114,115,116,117,118,119,120,121,122,123,124,125,126,127]
Mosaic algorithm HA	H1, H5 IAV	Modified Vaccinia Ankara (MVA) Plasmid DNA Recombinant Adenovirus	Mouse Non-human Primate	MVA delivery of H5 Mosaic protected mice from divergent homosubtypic challenge mediated through decreased lung inflammation and viral replication durable up to 6 months post-vaccination. T cells stimulated through H5 Mosaic vaccination possessed cross-reactivity with H1 C-terminus. Rhesus macaques vaccinated with H5 Mosaic produced antibodies against both H5 and H1 subtypes, which mediated protection through both HAI and ADCC activity post-challenge. H1 Mosaic vaccination induced broad cross-reactive antibody responses better than wild-type HA and commercial comparator vaccination. Vaccination with H1 Mosaic protected mice in dose-dependent manner.	[128,129,130]
Epigraph algorithm HA	H3 IAV, IBV	Recombinant Adenovirus	Mouse Swine Ferret	Epigraph vaccination induced durable cross-reactive HAI antibodies and IFN-γ^+^ T-cell responses in BALB/c mice and pigs. Epigraph vaccination protected mice from lethal and non-lethal human H3 challenge. Protection afforded by Epigraph vaccination led to lower infectious viruses in the lungs of pigs 6-months post-vaccination. Both CD4+ and CD8^+^ T cells play a role in protection from lethal challenge in mice, with some evidence of non-neutralizing antibodies mediating protection. Individual immunogens differentially contributed to immune response to human H3 Epigraph vaccination. Cross-reactive immune responses were observed in ferrets along with protection from clinical and microscopic disease pathology after challenge.	[131,132,133,134]

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
