# Peer review of "Influenza B Virus Vaccine Innovation through Computational Design"

_pathogens, 2024, doi:10.3390/pathogens13090755_

Round 1

Reviewer 1 Report

Comments and Suggestions for Authors

This manuscript by Weaver et al reviews the current status of  Influenza B virus vaccine design using computational approaches. This is a well organized and well written review article and provides a comprehensive overview of the available literature. 

Minor comments:

1. Please re-design/re-format the table to fit in one page.

2. Please include clinical studies for Influenza B virus vaccine.

3. Please discuss (in a few sentences) about any shorts falls of using computational approaches for vaccine design.

Reviewer 2 Report

Comments and Suggestions for Authors

This is well-written manuscript. The information presented is clear and logical and supported by appropriate and up-to-date references.  The manuscript provides comprehensive overview of current state of IAV/IBV spread and its recent evolutionary pattern and discusses various strategies of immunogen presentation being undertaken in the pre-clinical research along with various computational design strategies to develop effective IBV vaccines against fast mutating and emerging virus. I have no comments. 

Reviewer 3 Report

Comments and Suggestions for Authors

The present review summarized the various stages and development of vaccine for Influenza B Virus through in-silico design strategies. The topic is relevant in todays time. The article is well structured and presented. The references cited are appropriate. However, before the article can be considered for application, the following queries need to be addressed. 

1. In section 3, which disucsses about 155 Pre-Clinical Vaccine Research, it is important to add one figure/table to summarize the major regions/epitopes being targted and the challeneges therein. This would give the readers a snapshopt about the area before proceeding to the next section. 

2. In section 4, about computational design strategies, it is important to have separate section for vaccines being developed with multiple epitopes. 

3. The comparative efficacy of the vaccines in terms of delivery route can be highlighted. 

Round 2

Reviewer 3 Report

Comments and Suggestions for Authors

The authors have satisfactorily answered the queries raised during the revision. The addition of necessary calrifications in different sections and Figure 2 has made the revised version better to comprehend. The manuscript can now be considered for publication.